# Genome Sequencing and Transcriptome Profiling in Twins Discordant for Mayer-Rokitansky-Küster-Hauser Syndrome

**DOI:** 10.3390/jcm11195598

**Published:** 2022-09-23

**Authors:** Rebecca Buchert, Elisabeth Schenk, Thomas Hentrich, Nico Weber, Katharina Rall, Marc Sturm, Oliver Kohlbacher, André Koch, Olaf Riess, Sara Y. Brucker, Julia M. Schulze-Hentrich

**Affiliations:** 1Institute of Medical Genetics and Applied Genomics, University of Tübingen, 72076 Tübingen, Germany; 2Applied Bioinformatics, Department of Computer Science, University of Tübingen, 72076 Tübingen, Germany; 3Department of Women’s Health, University of Tübingen, 72076 Tübingen, Germany; 4Rare Disease Center Tübingen, University of Tübingen, 72076 Tübingen, Germany; 5Institute for Bioinformatics and Medical Informatics, University of Tübingen, 72076 Tübingen, Germany; 6Institute for Translational Bioinformatics, University Hospital Tübingen, 72076 Tübingen, Germany; 7Research Institute for Women’s Health, University of Tübingen, 72076 Tübingen, Germany

**Keywords:** MRKH syndrome, monozygotic discordant twins, genome sequencing, transcriptome analysis, Müllerian ducts

## Abstract

To identify potential genetic causes for Mayer-Rokitansky-Küster-Hauser syndrome (MRKH), we analyzed blood and rudimentary uterine tissue of 5 MRKH discordant monozygotic twin pairs. Assuming that a variant solely identified in the affected twin or affected tissue could cause the phenotype, we identified a mosaic variant in *ACTR3B* with high allele frequency in the affected tissue, low allele frequency in the blood of the affected twin, and almost absent in blood of the unaffected twin. Focusing on MRKH candidate genes, we detected a pathogenic variant in *GREB1L* in one twin pair and their unaffected mother showing a reduced phenotypic penetrance. Furthermore, two variants of unknown clinical significance in *PAX8* and *WNT9B* were identified. In addition, we conducted transcriptome analysis of affected tissue and observed perturbations largely similar to those in sporadic cases. These shared transcriptional changes were enriched for terms associated with estrogen and its receptors pointing at a role of estrogen in MRKH pathology. Our genome sequencing approach of blood and uterine tissue of discordant twins is the most extensive study performed on twins discordant for MRKH so far. As no clear pathogenic differences were detected, research to evaluate other regulatory layers are required to better understand the complex etiology of MRKH.

## 1. Introduction

Mayer-Rokitansky-Küster-Hauser (MRKH) syndrome is a rare congenital condition present in approximately one in 5000 live female births manifesting as aplasia or severe hypoplasia of structures related to the development of the Müllerian duct including the uterus and upper part of the vagina [1]. MKRH can either occur as an isolated form (type 1) or with additional extragenital phenotypes (type 2) such as renal or skeletal malformations [2]. Even though the occurrence of leiomyoma in women with MRKH is rare, multiple cases have been reported in the literature over the years, making it an important and easily overlooked differential diagnosis in MRKH patients presenting with abdominal pain [3,4]. Another difficulty women with MRKH face, is the inability to have sexual intercourse without previous neovagina surgery and to reproduce naturally. This can lead to severe psychological strain, further highlighting the importance of investigating this condition in more detail [5].

Females affected by MRKH syndrome usually have a normal 46.XX karyotype, normal ovarian function, and normal secondary sexual characteristics. In addition to various sporadic cases, several familial occurrences have been observed, suggesting a potential genetic origin of the syndrome [6]. Multiple genes have been subject to intensive investigation but could only partly be linked to a subset of MRKH cases in individual cohorts. Among them are genes essential for Müllerian and Wolffian duct development, which might play a key role in MRKH etiology [7]. Copy number variants (CNVs) in 1q21.1, 16p11.2, 17q12, and 22q11 with the corresponding genes *LHX1* and *HNF1B*, WNT signaling pathway genes such as *WNT4*, *WNT9B*, and the HOX family as well as some others have been pivotal candidates [8]. Recently, whole exome sequencing (WES) was used to identify *GREB1L* as a potential candidate for MRKH type 2 cases with kidney anomalies [9], but in general, the multifaceted and complex phenotype of the type 2 MRKH syndrome makes interpretation of genetic findings challenging. Until recently, previous work only focused on genome-wide CNV detection using array-CGH assays or WES and SNP arrays [10,11] limiting detection to exonic variants and therefore discarding potential mutations in regulatory factors. In 2019, Pan et al. [12] published results based on whole genome sequencing (WGS) for the detection of de novo variants in nine MRKH type 1 patients, which still remains, up to our knowledge, the only publication with a whole genome approach for MRKH patients complementing previously performed work.

As MRKH occurs mostly sporadically, this syndrome could also be explained by *de novo* dominant mutations in the germline of the offspring, or somatic mutations in the affected tissue. Those somatic variants, which might occur in early development, could also explain cases of monozygotic twins discordant for the MRKH syndrome [13,14,15]. Up until now, we are not aware of any publication analyzing and discussing tissue-specific variants in MRKH patients, which leads to certain unknowns concerning potential genetic mosaics in uterus rudiments as a potential source for MRKH type 1 and type 2 malformations.

To better understand the genetic contribution to MRKH etiology, we herein sequence and analyze the whole genome of blood and affected uterus rudiment of five twin pairs discordant for MRKH in an attempt to include regulatory features as well as potential tissue-specific gene alteration. In addition, we extend our tissue-specific endometrial transcriptome analysis from sporadic cases [16] to monozygotic discordant twins and evaluate perturbations in gene expression underlying MRKH.

## 2. Materials and Methods

### 2.1. Patients and Ethical Approval

From five MRKH-discordant pairs of MZ twins, blood samples, and clinical data were collected during routine clinical visits at the Department of Obstetrics and Gynecology, Tübingen University Hospital, Tübingen, Germany. The study was approved by the Institutional Review Board of the University of Tübingen (approval number: 205/2014BO1) and informed consent for genetic studies was obtained from each patient before recruitment.

Of the five patients with MRKH, one had type I and four had type II MRKH with associated renal malformations, one of whom had ureter abnormalities. Except for one pair, where both had renal agenesis, the corresponding co-twins were not affected by genital, renal, skeletal, or other malformations (Table 1). Tissue from uterine rudiments was collected from the affected twin during laparoscopic creation of a neovagina.

### 2.2. DNA Isolation and Sequencing

DNA from blood was isolated using the DNeasy purification kit and automated on the QIA-cube (Qiagen, Santa Clarita, CA, USA). DNA from rudimentary uterine tissue was isolated with the QiaAmp DNA Mini Kit (Qiagen). Genome sequencing was performed using the *TruSeq* PCR-free kit (Illumina, San Diego, CA, USA) on an Illumina platform (NovaSeq6000). Generated sequences were processed using the *megSAP* analysis pipeline (https://github.com/imgag/megSAP, accessed on 20 December 2021). *megSAP* performs quality control, read alignment, various alignment post-processing steps, variant detection, as well as comprehensive annotation of variants. Diagnostic analysis was performed using the *GSvar* clinical decision support system (https://github.com/imgag/ngs-bits, accessed on 20 December 2021). We used the GRCh38 reference genome assembly for the analysis.

### 2.3. Genome Analysis and Variant Calling

The resulting data were then analyzed implementing different filter criteria and sample constellations.

Initially, the blood and tissue samples of the affected twins were compared to the respective healthy twin’s blood samples using a multi-sample dominant filter (allele frequency < 0.10%, coding or near splice variants, keep pathogenic variants of HGMD or ClinVar, quality filter (quality: 250|depth: 0|mapq: 55|strand bias: 20|allele balance: 80)). We specifically searched for variants present in uterine tissue and blood of the affected twin, while absent in blood of the unaffected twin. Later, the multi-sample dominant filter was modified by including all coding or non-coding variants in MRKH target regions. This filter contained 612 regions and 752 different genes already described in the literature to be involved in uterine development and tissue genesis pathways.

The second analysis focused on comparing the affected twin’s tissue and blood using once more the modified multi-sample dominant filter and target region filter.

Lastly, we analyzed for rare (<1% allele frequency) single nucleotide variants (SNVs), CNVs, or structural variants in the MRKH target region in the affected twin’s tissue.

All remaining variants of the previous analyses were then manually evaluated, with respect to quality, conservation, the gene’s biological functions, and previous association to MRKH.

### 2.4. Validation Using Sanger Sequencing

To validate the variant found in *ACTR3B*, primer pairs for Sanger sequencing were designed with Geneious Prime (Dotmatics, San Diego, CA, USA), a platform utilizing the widely used Primer3 algorithm. The resulting forward- (*ACTR3B*-F: TACTCTCAGGAGGCTCCACC) and reverse-primer (*ACTR3B*-R: CCAGGGAGAGTGTGAAGCTG) were produced by metabion international AG (Steinkirchen, Germany). PCR was then performed using the Taq DNA Polymerase Kit (Qiagen, Santa Clarita, CA, USA) with a final volume of 40 µL containing 0.4 µL of Taq DNA Polymerase, 0.8 µL dNTPs, 4 µL QIAGEN PCR Buffer, 8 µL Q-Solution, 21.6 µL Ampuwa^®^ (Fresenius, Bad Homburg, Germany), 2 µL genomic DNA (c = 36 ng/µL (blood samples); 110 ng/µL (tissue sample)) and 1.6 µL of each primer (10 µM). The resulting PCR amplicons were purified using the QIAquick^®^ PCR Purification Kit (Qiagen, Santa Clarita, CA, USA) and later sequenced with the DCTS Quick Start Kit and CEQ™ 8000 Genetic Analysis System both by Beckman Coulter Inc. (Brea, CA, USA). All procedures were performed according to the manufacturer’s manual.

### 2.5. RNA Isolation and Sequencing

Total RNA from the endometrium of rudimentary uterine tissue was isolated using the RNeasy Mini Kit (Qiagen) and used for paired-end RNA-seq. Quality was assessed with an Agilent 2100 Bioanalyzer (Santa Clara, CA, USA). Samples with high RNA integrity numbers (RIN > 7) were selected for library construction. Using the NEBNext Ultra II Directional RNA Library Prep Kit for Illumina and 100 ng of total RNA for each sequencing library, poly(A) selected paired-end sequencing libraries (101 bp read length) were generated according to the manufacturer’s instructions. All libraries were sequenced on an Illumina NovaSeq 6000 platform at a depth of around 40 mio reads each. Library preparation and sequencing procedures were performed by the same individual, and a design aimed to minimize technical batch effects was chosen.

### 2.6. RNA Quality Control, Alignment, and Differential Expression Analysis

Read quality of RNA-seq data in fastq files was assessed using *FastQC* (v0.11.4) [17] to identify sequencing cycles with low average quality, adaptor contamination, or repetitive sequences from PCR amplification. Reads were aligned using *STAR* (v2.7.0a) [18] allowing gapped alignments to account for splicing against the *Ensembl* H. sapiens genome v95. Alignment quality was analyzed using *samtools* (v1.1) [19]. Normalized read counts for all genes were obtained using *DESeq2* (v1.26.0) [20]. Transcripts covered with less than 50 reads (median of all samples) were excluded from the analysis leaving 15,131 genes for determining differential expression. We set |log_2_ fold-change| ≥ 0.5 and BH-adjusted *p*-value ≤ 0.05 to call differentially expressed genes. Gene-level abundances were derived from *DESeq2* as normalized read counts and used for calculating the log_2_-transformed expression changes underlying the expression heatmaps for which ratios were computed against mean expression in control samples. The *sizeFactor*-normalized counts provided by *DESeq2* also went into calculating nRPKMs (normalized reads per kilobase per million total reads) as a measure of relative gene expression [21]. Upstream regulators as well as predicted interactions among DEGs were derived from *Ingenuity Pathway Analysis* (IPA, v01–16, Qiagen). *Cytoscape* was used for visualizing networks [22]. Cell type-specific endometrial marker genes are based on single-cell data [23].

## 3. Results

### 3.1. Case Reports

For our study, we recruited five pairs of discordant monozygotic (MZ) twins of which one sister was affected with MRKH while the other sister did not show vaginal and uterus aplasia (Table 1). We obtained EDTA blood from both twins, their parents, as well as unaffected siblings willing to participate. We were also able to obtain tissue from the uterine rudiment of the affected twin (except twin 3). Additionally, kidney malformations were observed in four affected individuals (1-1, 2-1, 3-1, and 5-1), as well as the twin sister 3-2 who did not show features of MRKH. Additional phenotypic features such as skeletal, heart, or eye malformations were observed in three affected individuals (1-1, 2-1, and 5-1). In addition to twin sisters 3-2, none of the twins or siblings showed any of these additional phenotypes.

### 3.2. Multi-Sample Analysis of Twin Genomes for Discordant Variants

In the first analysis step, we performed multi-sample analyses of the blood and tissue of the affected twin against the blood of the unaffected twin as a control (Appendix A). This filtering step gave us an average of 16 variants in the coding region, as well as seven variants in the non-coding regions. Since this filtering step enriched for technical artifacts, variant quality was assessed manually using IGV and resulted in only one mosaic variant of good quality in *ACTR3B* in twin 2 (Figure 1, Table 2). This variant (*ACTR3B* (ENST00000256001.13):c.1066G>A, p.Gly356Arg) had an approximate allele frequency of 39% in the affected tissue, 10% in the blood of the affected twin and only one read in the blood of the unaffected twin 2-2 (Figure 1). *ACTR3B* encodes a member of the actin-related proteins and plays a role in the organization of the actin cytoskeleton. It is highly expressed in the brain and many other tissues (https://www.proteinatlas.org/ENSG00000133627-ACTR3B/tissue, accessed on 15 February 2022). The glycine at position 365 is highly conserved and within the actin domain. Variants in *ACTR3B* have not been implied in MRKH or genitourinary malformation before and the functional consequences of this variant remain unclear.

For the other twin pairs, a multi-sample analysis of tissue against the blood of the affected twin did not reveal any discordant variants.

Furthermore, we performed analyses for copy number variants and structural variants only present in the affected twin. Unfortunately, no variant with sufficient quality could be identified.

### 3.3. Analysis of Genome Data for Rare Conserved Variants in MRKH Candidate Genes

Under the assumption that the MRKH phenotype might not be fully penetrant, genome data were further analyzed for rare conserved single nucleotides as well as copy number or structural variants (Appendix A). This revealed one pathogenic variant in twin 3-1 (affected) and twin 3-2 (renal agenesis without features of MRKH), carrying a stop variant in *GREB1L* (ENST00000269218.10):c.4665T>A, p.Tyr1555* (Table 2). *GREB1L* is a target gene in the retinoic acid signaling pathway, which is highly expressed in the developing fetal human kidney and involved in the early metanephros and genital development [24]. Dominant variants in *GREB1L* have been previously described for renal hypodysplasia/aplasia 3 (OMIM #617805) including uterine abnormalities and MRKH. Interestingly, the twin sister not affected with MRKH had unilateral renal agenesis. Segregation analysis showed that this variant was inherited from a healthy mother. Reduced penetrance is well known for *GREB1L* variants [24,25]; thus, we classified this variant as pathogenic.

Furthermore, we identified a heterozygous *PAX8* variant (ENST00000263334.9: c.1315G>A, p.Ala439Thr) in twin 1-1 (affected), twin 1-2 (unaffected), as well as the unaffected mother (Table 2). *PAX8* has been postulated in MRKH before [7]. The variant affects a well-conserved amino acid within the paired-box protein 2C terminal domain of PAX8 and is not listed in gnomAD. According to ACMG criteria, we classified this variant as a variant of unknown significance.

In twin 2-1 (affected) and twin 2-2 (unaffected), a missense variant in *WNT9B* (ENST00000290015.7:c.205C>T, p.Arg69Trp) was detected which is also present in the unaffected sister and was inherited from the father (Table 2). Variants in *WNT9B* have been implicated in MRKH before [26,27]. This variant affects a conserved amino acid within the WNT domain and is listed in gnomAD [28] with a frequency of 0.0002. Among the individuals carrying this variant in gnomAD, there is no shift between female and male carriers. Nevertheless, we cannot exclude an effect of this variant on the MRKH phenotype potentially in combination with other variants. Thus, we classified this variant as variant of unknown significance.

### 3.4. MRKH Twins and Sporadic Cases Showed Largely Similar Endometrial Transcriptome Changes

To test whether the observed variants in *ACTR3B*, *GREB1L*, *PAX8*, and *WNT9B* altered their gene expression levels, we performed RNA-sequencing of uterine rudiments obtained from three affected twins (1-1, 2-1, and 4-1). Compared to previously published controls and sporadic MRKH patients [16], no significant change in gene expression was observed for these genes (Figure 2), indicating that the variants did not alter the underlying expression.

As discordant MRKH twins might be based on a distinct etiology when compared to sporadic cases, we next asked whether the overall endometrial transcriptome differs between discordant twins and sporadic cases. Therefore, we compared the three twin transcriptomes to previously published transcriptomes of 35 sporadic patients (19 type 1 and 16 type 2) as well as 25 controls [16]. Principal component analysis separated MRKH samples from controls and the three twin samples clustered together with those of sporadic cases, indicating a high similarity of transcriptomic signatures (Appendix A). Based on single-cell data from endometrial tissue [23], cell type-specific expression of marker genes from ciliated and unciliated epithelial cells was highly similar between twins and sporadic cases (Appendix A).

To compare these commonalities in more detail, differential expression changes were determined between MRKH twins and control samples. When compared to controls, a total of 449 differentially expressed genes (DEGs) with 235 being up- and 214 being down-regulated were identified (Figure 3A). Of those, 174 were also significantly changed in MRKH sporadic cases (Figure 3B), which in total included 2121 DEGs as previously reported [16]. Interestingly, 26 of the 174 DEGs showed opposing expression changes between twins and sporadic cases with the MZT twins appearing to be more similar to the expression pattern in controls than sporadic cases (Figure 3C, Appendix A). These 26 DEGs were enriched for the Gene Ontology term *hydrolase activity* including genes such as *ATAD3B*, *CHTF18*, *HAGHL*, *HDAC10*, *PLA2G6*, and *RTEL1-TNFRSF6B*. However, the majority of the shared 174 DEGs showed largely similar expression changes in twins as well as sporadic samples when compared to controls (Figure 3C).

To better understand the underlying biology of the endometrial perturbances in MRKH twins, enrichment analysis was then applied to identify molecules upstream of the observed DEGs. The most significant molecule observed was *fulvestrant* (Figure 4A), a selective estrogen receptor degrader (SERD), used as a medication to treat hormone receptor (HR)-positive metastatic breast cancer in postmenopausal women with disease progression as well as HR-positive, HER2-negative advanced breast cancer in combination with *palbociclib* in women with disease progression after endocrine therapy [29,30]. In line, the estrogen receptor itself was found as a significant upstream regulator (Figure 4A). Visualizing a network of DEGs around *fulvestrant* connects several key genes such as *WNT4* previously linked to MRKH (Figure 4B). Individually plotting gene expression changes of these candidates, further points to the similarity of endometrial gene expression changes between MRKH twins and sporadic cases (Figure 4C).

Taken together, our transcriptome analysis indicates that MRKH twins do not substantially differ from other MRKH cases and that, in general, observations made in twins can be transferred to sporadic cases as well.

## 4. Discussion

### 4.1. Etiology of MRKH Syndrome

The etiology of MRKH syndrome is complex, multifactorial and, to date, not well understood. Even though a number of candidate genes have been proposed, many lack functional evidence. Additionally, many familial cases show an autosomal dominant inheritance with incomplete penetrance and variable expressivity [6,31], which complicates genetic diagnosis using segregation analysis. Especially, missense variants in candidate genes are very hard to interpret in this context but also nonsense variants pose difficulties in this regard. While some studies showed that pathogenic loss of function variants in candidate genes such as *PAX8*, *BMP4,* and *BMP7* are predominantly inherited by the father [7], there is no sex difference seen for loss of function variants in gnomAD for these genes. This points towards reduced penetrance and further mechanisms involved in MRKH. Studying discordant twins poses an opportunity to investigate variants differing between the twins as well as looking for potential epigenetic marks and transcriptomic changes between these twins.

### 4.2. Genomic Differences of Monozygotic Twins

Even though monozygotic twins developed from the same zygote, they do not have the exact same genome. During development, a number of variants occur and may differ in variant allele frequency (VAF) between both twins depending on the timepoint and cell population of twinning. Some of these variants may be nearly constitutional with a VAF of >0.45, while others are mosaic with lower VAFs. A recent study of 381 monozygotic twins and two triplets estimated the average number of early developmental variants differing in twins to be around 5.2 [32]. The number of discordant variants observed for each twin pair had a huge variation ranging from more than 100 to no variant observed. In our study, we could only identify one mosaic variant in a total of five twin pairs and no nearly constitutional variant differing in the twins. One explanation could be that our cohort size is much smaller and may be biased towards a lower number of discordant variants. Another explanation could be that while we had average coverage of about 43×, the study by Jonsson and colleagues had coverage of 152× for many of their genomes leading to higher accuracy of variant and mosaic calling.

In a previous study, the same monozygotic twin pairs were analyzed using SNP arrays and potential CNVs of unknown clinical relevance were identified in affected twin 5-1 but not in her healthy sister [33]. These CNVs could not be replicated in this study. One reason for this could be the different methods applied. Recently, a study found that arrays sometimes produce artifacts, and genome data are more reliable for CNV calling (https://www.biorxiv.org/content/10.1101/2022.06.10.495642v1.full). Another reason for this discrepancy could be mosaic frequencies caused by analyzing DNA that was isolated from different parts of the affected tissue.

### 4.3. Mosaic Variant in ACTR3B

The only discordant variant we could identify in our twin cohort was a mosaic variant c.1066G>A, p.Gly356Arg in *ACTR3B*. *ACTR3B* encodes for an actin-related protein (ARP) involved in the organization of the cytoskeleton which has not been implied in MRKH or any other disorder so far. GTEx expression data for several tissues show a predominant expression of *ACTR3B* in the brain and a rather low expression in the uterus (https://www.proteinatlas.org/ENSG00000133627-ACTR3B/tissue, accessed on 15 February 2022). In contrast, protein levels of ACTR3B are relatively high in female tissue. During murine development, it is mainly expressed in embryonal ectoderm and not mesoderm or endoderm (derived from GXD [34]), which are more crucial to the development of the genital tract. ACTR3B is highly intolerant to loss of function variants with a pLI-score of 1 and the affected amino acid Gly356 is highly conserved within the actin domain. Still, without further functional evidence and the identification of further individuals with variants in *ACTR3B* and genitourinary anomalies, the connection between this mosaic variant and the observed phenotype remains elusive.

### 4.4. Pathogenic GREB1L Variant

Our analysis for rare conserved variants in candidate genes for MRKH identified a heterozygous loss of function variant c.4665T>A, p.Tyr1555* in *GREB1L*. *GREB1L* has been previously associated with a variety of genitourinary disorders (OMIM #617805) including MRKH [9,35,36,37]. Many affected families present with a pedigree with autosomal dominant inheritance and reduced penetrance and variable expressivity. This is also seen in the family presented here. The mother shows no symptoms of genitourinary malformations while one daughter has MRKH and a pelvic kidney (MRKH type 2) and her twin sister has unilateral kidney agenesis, hip dysplasia, and scoliosis with no evidence of MRKH. In previous reports on pathogenic *GREB1L* variants, the authors noted a bias in transmission towards maternal inheritance which might be caused by imprinting of the paternal allele [24,36]. The mothers transmitting the variant were usually unaffected or only showed a mild phenotype. This could also explain why the mother of twin pair 3 is unaffected but both her daughters show symptoms of *GREB1L* haploinsufficiency.

### 4.5. Variants of Unknown Significance

#### 4.5.1. *WNT9B*

*WNT9B* encodes a secretory glycoprotein and paracrine factor that is part of the canonical Wnt signaling pathway, one of the main pathways responsible for the organization of the mammalian urogenital system by regulating embryonic development, adult tissue homeostasis, and differentiation [38]. In mice, it is mostly expressed in the developing kidney and epithelium of the Wolffian duct in both sexes, therefore playing an essential role in renal development [38].

Even though the gene is not highly expressed in the Müllerian duct, which later forms the uterus, oviduct, and upper part of the vagina, posterior Müllerian duct elongation is only made possible by Wnt9b mediated interactions with the epithelium of the Wolffian duct [38,39]. A study with *Wnt9b−/−* knock-out mice corroborates the above-mentioned mechanism, as males did not develop a vas deferens and epididymis and females lacked a uterus, oviduct, and upper vagina, whereas the gonads appeared normal, as is the case in MRKH [38].

Multiple studies found that some patients with Müllerian duct anomalies or MRKH did indeed have variants in *WNT9B* [26,27]. Furthermore, SNPs in this gene were linked to a higher MRKH risk [40]. However, a different study, comparing *WNT9B* variants of patients with Müllerian abnormalities to healthy controls, ruled out *WNT9B* variants as the only causative factor for Müllerian anomalies, as most of them could be found in a small number of healthy controls as well [41]. This phenomenon could also be observed in twin pair 2, which implies that WNT9B is not the only causative factor of MRKH, but might still be involved in its, yet to be understood, pathogenesis.

#### 4.5.2. *PAX8*

The *PAX8* gene encodes a homeodomain signaling molecule, strongly expressed in the Müllerian duct, Wolffian duct, and kidney during embryonic development [42]. Together with *PAX2*, a gene closely related and partly redundant in function to *PAX8*, they initiate the mesenchymal–epithelial transition which later leads to the formation of the Wolffian duct [43]. The same mechanism is believed to be involved in the Müllerian duct formation [39]. Animal models corroborate this theory as *Pax2−/−* mice do not have kidneys nor a reproductive tract [43], only initially developing the anterior part of the Wolffian and Müllerian duct [44]. In *Pax2−/−Pax8−/−* embryos there is no Wolffian nor Müllerian duct formation at all.

The studies on *Pax8−/−* knock-out mice are quite controversial. Whereas Mittag et al. observed only remnants of myometrial tissue instead of a uterus and the absence of a vaginal opening in *Pax8−/−* mice [45], Mansouri et al. reported no urogenital malformations with the same knockout model [42]. Even if it is not yet clear whether a variant in *PAX8* alone is enough to disrupt urogenital development, the gene is certainly involved in this delicate process.

Furthermore, multiple *PAX8* variants and deletions have already been described in MRKH patients [7,46], indicating a connection between this gene and the MRKH pathogenesis.

### 4.6. Transcriptome Analysis of MRKH Twins

In addition to our genome analysis, we also used the chance to better understand gene expression perturbations in monozygotic twins and profiled the endometrial transcriptome of the affected twins. In contrast to a small subgroup of genes with opposing expression changes between twins and sporadic cases, the observed perturbations largely agreed with changes previously described in sporadic cases [16]. The main chemical associated with the shared DEGs was *fulvestrant*, a selective estrogen receptor degrader (SERD), used as a medication to treat hormone receptor (HR)-positive metastatic breast cancer in postmenopausal women with disease progression as well as HR-positive, HER2-negative advanced breast cancer in combination with *palbociclib* in women with disease progression after endocrine therapy [29,30]. In addition, the estrogen receptor itself was found as a significant upstream regulator and the gene encoding for estrogen receptor 1, *ESR1,* was perturbed in affected twins and sporadic cases in line with our previous findings [47,48,49].

Agreeing with previous literature, estrogens are necessary for the embryonic development of the female reproductive tract. The pivotal role of *ESR1* in female reproductive tract development has been previously shown by disrupting the corresponding gene in mice, subsequently leading to hypoplastic uterine and vaginal tissue [50]. Intriguingly, prenatal exposures of fetuses to synthetic estrogen such as diethylstilbestrol (DES) can also alter HOX gene expression in the developing Müllerian system, thereby disrupting the development of the female reproductive tract [51]. In the 1970s, DES was prescribed to pregnant women to prevent miscarriages until it was realized that daughters exposed to DES *in utero* showed a higher incidence of Müllerian anomalies [52] and a higher prevalence of MRKH [53].

Taken together, our transcriptome analysis confirms previous transcriptome perturbations in MRKH and points to high similarity of endometrial gene expression changes between twins and sporadic cases.

## 5. Conclusions

In this study, we examined the genetic differences of discordant twins as well as blood and affected tissue for MRKH using genome sequencing. To our knowledge, no previous study conducted genome sequencing for discordant MRKH twins or on the affected tissue. Even though our intention was to shed light on the genetic causes of MRKH by thoroughly checking for small copy number variants or structural variants missed by previous array analysis as well as identifying potential pathogenic variants in intronic, intergenic regions and non-coding transcripts, we were not able to identify any such causes neither in tissue nor as discordant variants nor as variants in candidate regions. One explanation for this could be the problem of interpreting such variants. Still, with our approach of sequencing discordant twins and more than one tissue, we were not able to identify a discordant variant in non-coding regions, indicating that it is more complex than just the interpretation of the variants in question. Another possibility would be that even after decades of deciphering the genome, some genomic regions are still not correctly mapped due to their highly repetitive nature. With our short-read genomes, we would not have been able to identify changes in these regions but would rather need long-read genome sequencing. Furthermore, our findings support the claim that MRKH is likely not inherited in a dominant fashion, which has also been observed in a study evaluating the frequency of congenital anomalies in biological children of MRKH patients [54]. The aforementioned study found that none of the 17 examined female children of MRKH mothers inherited the syndrome [54]. To summarize, MRKH patients do not appear to directly pass on the congenital condition to their children and as a general genetic cause has, to our knowledge, not yet been identified, the risk of MRKH occurring in biological daughters of affected women is small. The only clearly pathogenic variant we were able to identify was a loss of function variant in *GREB1L*. This shows that most likely previous studies of exome sequencing of blood did not miss out on a significant portion of pathogenic genetic variants. One explanation could be that the etiology of MRKH is far more complex, and incomplete penetrance, as seen for example in TAR syndrome [55], and variable expressivity hamper the identification of causative variants. The phenotypic discordance in the twins could also be explained by possible tissue mosaicisms not detected in this study. Another reason is that genetics might play a minor role in the development of MRKH and epigenetic changes as well as environmental and microbiological factors should be taken more into account when looking into the etiology of MRKH.

## Figures and Tables

**Figure 1 jcm-11-05598-f001:**
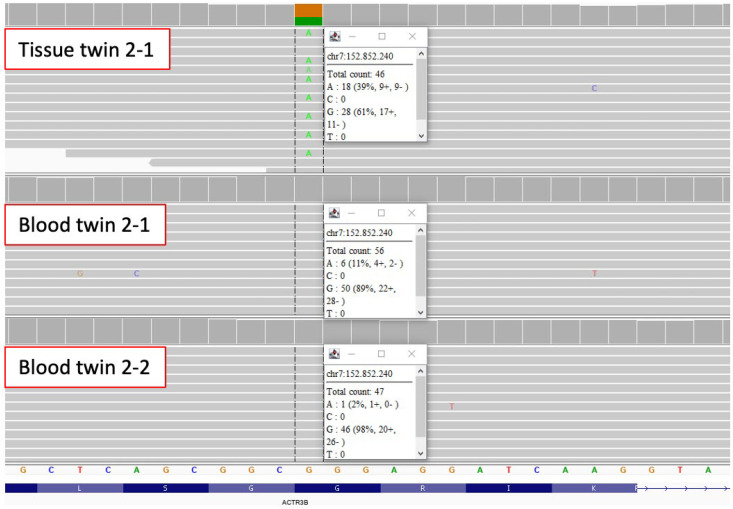
Mosaic missense variant in *ACTR3B* identified in blood and tissue of twin 2-1. IGV reads show that the variant c.1066G>A, p.Gly356Arg in *ACTR3B* has an allele frequency of about 39% in uterine tissue and an allele frequency of about 11% in blood of the affected twin 2-1, while blood of twin 2-2 only has 1 supporting read.

**Figure 2 jcm-11-05598-f002:**
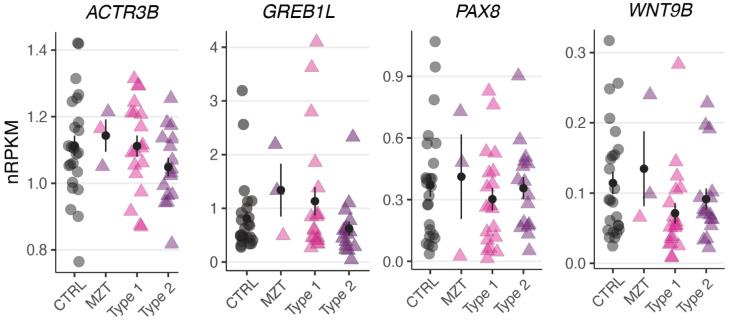
No significant change in genes expression observed for *ACTR3B*, *GREB1L*, *PAX8*, and *WNT9B*. Expression levels for the selected DEGs plotted as individual data points with mean ± SEM.

**Figure 3 jcm-11-05598-f003:**
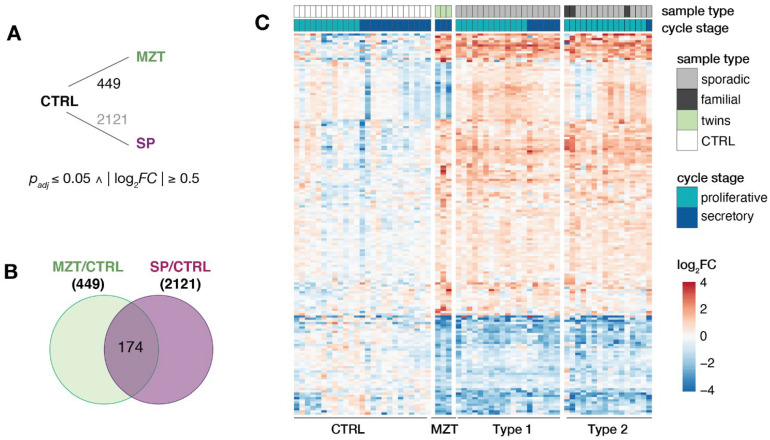
Endometrial transcriptomes of MRKH twins and sporadic cases show largely similar changes. (**A**) Schematic diagram of comparisons between MRKH monozygotic twins (MZT), MRKH sporadic cases (SP) and unaffected women (controls) indicating number of differentially expressed genes (DEGs). DEGs between sporadic cases and controls based on previous work [13]. Fold change and significance cut-offs below. (**B**) Venn diagram showing number of common DEGs MRKH twins and sporadic cases each compared to controls. (**C**) Expression profiles (*log_2_* expression change relative to *Ctrl* group) of 174 DEGs (common DEGs indicated in Figure 3B) across all samples. Rows hierarchically clustered by Euclidian distance and *ward.D2* method. Cycle information (proliferative or secretory) and patient type (monozygotic twin, sporadic, or control) on top.

**Figure 4 jcm-11-05598-f004:**
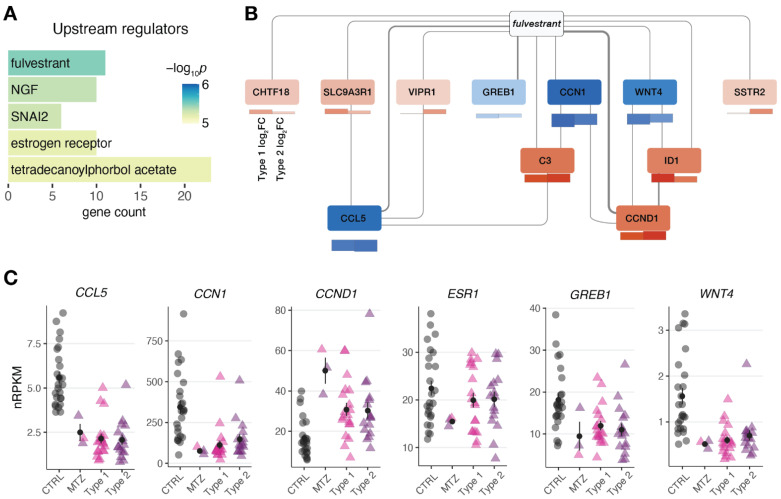
Gene expression changes in MRKH monozygotic twins point at regulators linked to estrogen receptor. (**A**) Predicted upstream regulators for common 174 DEGs (from Figure 3B) based on *Ingenuity Pathway Analysis*. Top five significant regulators shown. (**B**) Network of DEGs associated with upstream regulator *fulvestrant*. Interactions based on Ingenuity Pathway Analysis with line width indicating number of curated interactions. Genes color-coded by mean expression change observed in MZT/CTRL on top and SP/CTRL separately for type 1 and type 2 below. (**C**) Expression levels for the selected DEGs plotted as individual data points with mean ± SEM.

**Table 1 jcm-11-05598-t001:** Phenotype of individuals in this study.

Twin Pair	Individual	Age at Surgery (Years)	Mrkh Type	Kidney Malformation	Skeletal Malformation	Heart Malfomation	Vision
1	1-1	29	MRKH 2	Malrotation of kidney	Herniated disc	Fallot tetralogy	
2	2-1	19	MRKH 2	Kidney agenesis			Strabism
3	3-1	32	MRKH 2	Pelvic kidney			
3	3-2	-	-	Kidney agenesis	Hip dysplasia, scoliosis		
4	4-1	19	MRKH 1	-			
5	5-1	16	MRKH 2	Kidney agenesis			Strabism, poor vision

**Table 2 jcm-11-05598-t002:** Identified variants in SNV analysis.

Analysis Type	Sample	MRKHS Classification	Zygosity	Transcript (ENST-Number)	Variant Reads	Total Reads	Gene	cDNA Change	Protein Change	phyloP	gnomAD-Allele Frequency	Inheritance
Single sample analysis	Tissue MRKH-twin 1	MRKH-I	het	ENST00000263334.9	14	37	PAX8	c.1315G>A	p.Ala439Thr	42.050	0	AD
Single sample analysis	Tissue MRKH-twin 2	MRKH-I	het	ENST00000290015.7	20	31	Wnt9B	c.205C>T	p.Arg69Trp	27.030	0.0002180	AD
Single sample analysis	Tissue MRKH-twin 3	MRKH-II	het	ENST00000269218.10	33	51	GREB1L	c.4665T>A	p.Tyr1555Ter	-0.7730	0	AD
Multi sample analysis	Tissue MRKH-twin 2	MRKH-I	het	ENST00000256001.13	18	46	ACTR3B	c.1066G>A	p.Gly356Arg	75.720	0	AD
	Blood MRKH-twin 2	MRKH-I	het		6	56						
	Blood healthy-twin 2	MRKH-I	wt		1	47	-	-	-	-	-	-

This table illustrates the SNVs found in the various analyses. The abbreveations are as follows: AD—autosomal dominant; het—heterozygous; n/a—not annotated; wt—wild type.

## Data Availability

Genome and RNA-sequencing data that support the findings of this study have been deposited in the European Genome-phenome Archive (EGA) (primary accession number: EGAS00001006370, EGAS00001006371).

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
