# Peer review of "Genome Sequencing and Transcriptome Profiling in Twins Discordant for Mayer-Rokitansky-Küster-Hauser Syndrome"

_jcm, 2022, doi:10.3390/jcm11195598_

Round 1

Reviewer 1 Report

I believe the manuscript is well written and adds some novelty to this as yet unknown syndrome. I only suggest enhancing the introduction by mentioning some clinical aspects and potential complications associated with the disease, in order to stimulate the interest of a wider medical population. (doi: 10.48095/cccg2021194; doi:10.3390/ijerph18115895;  doi: 10.5468/ogs.2019.62.4.294). 

The article is really interesting and well structured, correctly highlighting how much there is still to be clarified from a genetic point of view.

Author Response

We highly appreciate the comments of the reviewer and tried to adapt the manuscript accordingly.

The main point was to enhance the introduction by highlighting some clinical aspects and potential complications associated with the disease, in order to stimulate the interest of a wider medical population. We agree with the reviewer and added in line 41 ff:

Even though the occurrence of leiomyoma in women with MRKH is rare, multiple cases have been reported in literature over the years making it an important and easily overlooked differential diagnosis in MRKH patients presenting with abdominal pain (3, 4). Another difficulty women with MRKH face, is the inability to have sexual intercourse without previous neovagina surgery and to reproduce naturally. This can lead to severe psychological strain, further highlighting the importance of investigating this disease in more detail (5).”

Reviewer 2 Report

This investigation was planned to discover new data about very controversial issue, genetic-related MRHK. Previously there were not demonstrated any unambiguous genetic drivers for MRHK patients and this disease is seemed to be polyetiological. Unfortunately, the only one gene which the authors revealed to be mutated was GREB1L and this loss of function mutation does not a trigger for MRHK most likely. 

The references are rather fresh (40% not older than 5 years) although the authors do not cite the paper Rall, K. et al. Mayer-Rokitansky-Küster-Hauser syndrome discordance in monozygotic twins: matrix metalloproteinase 14, low-density lipoprotein receptor–related protein 10, extracellular matrix, and neoangiogenesis genes identified as candidate genes in a tissue-specific mosaicism. Fertility and Sterility,2015, 103(2), 494–502.e3.  At the same time this paper is also about genetic alterations in discordant monozygotic twins with MRKH and revealed some candidate genes which can be potentially related to MRKH. I recommend to discuss K. Rall et al. results and compare them to the authors ones.

Unfortunately, the results of endometrial transcriptomics is in Supplement which I do not have an access to. Nevertheless, I suppose that these data should be placed into a main body of the paper because even clinicians are dramatically interested in endometrial biology in MRHK patients due to their reproductive functions and fertility.

Author Response

We thank the reviewer for the comments and agree with the proposed changes.

As main point, the reviewer recommends to discuss K. Rall et al., 2015 results and compare them to the authors ones. This suggestion is highly valuable and we added a paragraph in the discussion (line 310 ff):

 “In a previous study, the same monozygotic twin pairs were analyzed using SNP arrays and potential CNVs of unknown clinical relevance were identified in affected twin 5-1 but not in her healthy sister (33). These CNVs could not be replicated in this study. One reason for this could be the different methods applied. Recently, a study found that arrays sometimes produce artefacts and genome data is more reliable for CNV calling (https://www.biorxiv.org/content/10.1101/2022.06.10.495642v1.full).  Another reason for this discrepancy could be mosaic frequencies caused by analyzing DNA that was isolated from different parts of the diseased tissue.

We further agree with the reviewer that clinicians are interested in endometrial biology of MRHK patients due to their reproductive functions and fertility and followed the suggestions to move supplementary data to the main body. In our opinion that makes most sense for Supplementary 1 showing expression for genes with observed variants (ACTR3B, GREB1L, PAX8, and WNT9B). This is now main Figure 2. The other two supplementary figures with a principal component and cell type composition analysis indicate the quality and homogeneity of the data and are meaningful as supplement.

Author Response

We thank the 3rd reviewer for the detailed report and suggestion how to improve the manuscript. The reviewer points out that it would have been interesting to see if there were variants in non-MRKH candidate genes in the large structural variants associated with MRKH. Indeed, we performed a multi-sample analysis across the entire genome without focusing on candidate genes (see paragraph 3.2 Multi-sample analysis of twin genomes for discordant variants in line 186 ff). With that approach we identified a mosaic variant in ACTR3B which we describe in detail. Furthermore, we performed analysis for copy number variants and structural var-iants only present in the affected twin. As indicated in line 204, no variant with sufficient quality could be identified.

The reviewer also points the attention to the ~35 genes in Figure 2C (now Figure 3C) that show different expression signatures from most of the sporadic and familial cases and asks what these genes are and whether there is a gene networks involved. We appreciate the comment and investigated these genes more closely. From the 174 common genes, 26 show opposing expression between twins and sporadic/familial cases. Based on gProfiler, these 26 genes are enriched for the GO term Hydrolase Activity(including the genes ATAD3B, CHTF18, HAGHL, HDAC10, PLA2G6, RTEL1-TNFRSF6B, ENSG00000258461, ENSG00000260729). We added this information to the results section (see line 257 ff) and added a new supplementary figure 3 to show the names of these 26 genes. In addition, we also attenuated our statement in the discussion that expression changes are similar between twins and sporadic cases (see line 398ff).

Furthermore, we agree with the reviewer that our sample size is too small to infer that WES will capture most mutations. Indeed, one needs hundreds, if not thousands of participants to make this statement. Hence, we are careful with major conclusions and try to avoid overstating the results.

With respect to the minor points.

-We thank the reviewer for the important idea in the introduction to use a less stigmatizing term such as congenital condition instead of disease. We changed the manuscript accordingly.

-The reviewer suggests point out in detail which genes correspond to each large chromosomal variant. We feel that this would exceed the frame of the introduction and are convinced that the reference to previous literature is more meaningful.

-Yes, that is a good idea. We spelled out whole exome sequencing (WES) and further below whole genome sequencing (WGS).

-Indeed, the word complex is not ideal in that context. We used the word challenging instead and changed the sentence to :”…but in general the multifaceted and complex phenotype of the type 2 MRKH syndrome makes interpretation of genetic findings challenging.”

-To clarify our results we used the twins’ nomenclature consistently and adapted the text accordingly (see lines 210-212; 220-222; 226; 239)

-The word uterine rudiment has been commonly used by us and others before and we would like to keep it for consistency.

-Yes, we entirely agree that a subgroup of genes in Figure 3C (previously 2C) behaves differently. As described above, we now added a new supplementary figure to highlight them and also analyzed their enrichment for a pathway separately (see line 257 ff).

-The cluster of genes for which expression differs between twins and sporadic cases is not enriched for the upstream regulator fulvestrant.

Thanks also for the grammar suggestions, we tried to correct accordingly and highlighted changes in yellow.

-We changed the structure of the sentences in lines 109-114 to convey their content more clearly

-We also changed the sentence structure in lines 179-181, in addition to adding a comma

-We changed the sentence at 312-3.

Reviewer 4 Report

The article is excellent and very interesting for the readers. The methodology is well presented and quite complex.

I would ask the authors to add some sentences at the very end of Conclusion section regarding clinical background of the study. Why this genetic science regarding MRKH syndrome  is important for clinicians? Can authors offer any possible ways to prevent MRKH syndrome base on their results?

In my opinion these are minor changes that should include some extra clinical information at the end of Conclusion section.

Author Response

We highly appreciate the comment of the 4th reviewer to add a section at the very end of the Conclusion with respect to clinical background of the study. Accordingly, we added a paragraph to the manuscript (see line 429ff) which reads:

Thus, our findings support the claim that MRKH is likely not inherited in a dominant fashion, which has also been observed in a study evaluating the frequency of congenital anomalies in biological children of MRKH patients (54). The aforementioned study found that none of the 17 examined female children of MRKH mothers inherited the syndrome (54). To summarize, MRKH patients do not appear to directly pass on the congenital condition to their children and as a general genetic cause has, to our knowledge, not yet been identified, the risk of MRKH occurring in biological daughters of affected women is small.”